# Cigarette Smoking in Response to COVID-19: Examining Co-Morbid Medical Conditions and Risk Perceptions

**DOI:** 10.3390/ijerph19148239

**Published:** 2022-07-06

**Authors:** Lisa M. Fucito, Krysten W. Bold, Sydney Cannon, Alison Serrantino, Rebecca Marrero, Stephanie S. O’Malley

**Affiliations:** 1Department of Psychiatry, Yale University School of Medicine, New Haven, CT 06511, USA; krysten.bold@yale.edu (K.W.B.); sydneyycannonn123@gmail.com (S.C.); alison.serrantino@yale.edu (A.S.); rebecca.marrero@yale.edu (R.M.); stephanie.omalley@yale.edu (S.S.O.); 2Yale Cancer Center, Yale University School of Medicine, New Haven, CT 06520, USA; 3Smilow Cancer Hospital at Yale-New Haven, New Haven, CT 06519, USA

**Keywords:** coronavirus pandemic, COVID-19, smoking, medical condition, comorbidity, age

## Abstract

During the initial wave of the Coronavirus Disease 2019 (COVID-19) pandemic in the U.S., information was mixed about the relative COVID-19 risks and potential benefits associated with cigarette smoking. Therefore, we sought to understand individual differences in the impact of COVID-19 on cigarette smoking in a sample of adults who reported recent use, with a particular focus on chronic medical conditions likely associated with increased COVID-19 risk. Participants completed an online survey of smoking behavior, demographic variables, medical history, and COVID-19 risk perceptions between July and August 2020 (N = 286). We examined whether medical conditions, COVID-19 risk perceptions and/or demographic characteristics were related to smoking changes in response to the pandemic (i.e., no change, decrease, increase) using multinomial logistical regression. Younger age, higher COVID-19 risk perceptions and Black versus White race were associated with greater odds of decreased smoking compared to no smoking change. Moreover, having at least one chronic medical condition was associated with greater odds of increased smoking relative to no change. The results have important implications for tobacco cessation treatment and preventive healthcare during the ongoing COVID-19 pandemic and other public health threats.

## 1. Introduction

The Coronavirus Disease 2019 (COVID-19) pandemic and associated broad social distancing and/or lockdown policies to curb its spread [1] had immediate personal and economic consequences [2,3,4,5,6]. These consequences, in turn, may have impacted individuals’ health-related behaviors, such as tobacco use. Early on during the pandemic, there were conflicting data on the relation between tobacco use and COVID-19 susceptibility and illness severity, with some initial reports suggesting that nicotine/cigarette smoking might reduce risk of severe illness [7,8,9,10,11]. The results of more recent systematic reviews and meta-analyses, however, provide more definitive evidence that having a cigarette smoking history appears to increase the risk of severe COVID-19 illness and poor outcomes [12,13,14,15]. Likewise, cigarette smoking may also increase vulnerability to developing post-COVID syndrome (i.e., prolonged illness following acute COVID-19 infection) [16]. A growing body of literature suggests that the initial waves of the pandemic impacted tobacco use patterns in varied ways [17]. For instance, some adults endorsed increased tobacco use due to greater stress/negative effect [18,19,20,21,22,23,24,25]. Conversely, others reported decreased use because of reduced ability to afford or access tobacco products [26,27], lower social opportunities where use typically occurred [28] and/or health concerns [20,29,30,31,32].

Despite these initial findings related to smoking behavior, important knowledge gaps remain. No studies to our knowledge have examined factors such as individuals’ medical history that may impact tobacco use during a pandemic. It is unknown whether those with medical conditions perceived greater COVID-19 risks from their tobacco use and, accordingly, were more likely to reduce their use. These questions are important for optimal tailoring and targeting of tobacco interventions and preventive healthcare, particularly as the COVID-19 pandemic persists, and some patients recovered from acute illness transition to more chronic COVID-19 syndromes [16]. To add to this evidence base and address these gaps, we conducted an online, cross-sectional survey in a representative sample of individuals who reported current smoking. The survey was conducted between July and August 2020, approximately four months after COVID-19 hit the U.S. We examined participants’ smoking behavior in response to the COVID-19 pandemic with a particular focus on the potential relations with co-morbid medical conditions and COVID-19 risk perceptions. The benefits of this survey are the ability to examine these associations at the broader population level and to generate evidence to inform health promotion and disease prevention strategies for COVID-19 and other important public health threats. 

## 2. Materials and Methods

### 2.1. Procedures

This study involved an anonymous, online Qualtrics survey conducted over four weeks, starting 16 July 2020 to 17 August 2020. At the start of the survey, volunteers were advised of the study purpose and asked to provide consent for participation. Upon providing informed consent, the survey then asked individuals to self-report demographic information (i.e., age, gender, race, ethnicity) and whether they had used cigarettes in the last 90 days to determine eligibility. If a volunteer reported being at least 18 years old and indicated “yes” to recent cigarette smoking, they were administered the remaining survey questions. The final survey took approximately 5–10 min to complete.

Participants were recruited through paid advertisements on Facebook and Instagram. Seven out of ten U.S. adults report using Facebook, and most report daily use [33]. One in four adults report using Instagram, but its use is greater among Hispanic and Black adults compared to White adults [33]. Among older adults (i.e., aged 65+), 50% use Facebook, which represents an increase of 30% within the past decade [33]. Facebook/Instagram advertising facilitates targeted recruitment strategies by allowing researchers to specify user demographic and geographic characteristics and Web search terms (i.e., “Interests”) for which to direct ads. A systematic review of Facebook advertising for health research demonstrated that Facebook was effective for recruiting representative samples, including hard-to-reach populations (e.g., individuals who report alcohol/substance use), and outperformed traditional recruitment methods (e.g., flyers, brochures) and other social media advertising [34]. Likewise, studies that compared online recruitment strategies that “push out” ad content to users engaged in other activities (i.e., paid Facebook and Google advertising) with online recruitment strategies that “pull in” users looking for paid work (i.e., Amazon Mechanical Turk (MTurk), Qualtrics, Craigslist) found that push out methods yielded more diverse samples [35,36]. 

We ran separate Facebook/Instagram ads, geographically restricted to the U.S., in batches to target specific age groups: (1) 18–30; (2) 31–40; (3) 41–50; (4) 51–60; (5) ≥61. We ran ads longer for the last two age categories to oversample older participants for whom medical co-morbidities are more common.

Eligible participants were ≥18 years of age, lived in the United States and reported use of combustible cigarettes within the past 90 days. We utilized various safeguards to ensure the quality and integrity of the data. Qualtrics has built-in fraud detection procedures including bot detection via a reCAPTCHA system and anti-ballot stuffing to prevent duplicate entries. In addition to these features, the following screening procedures were implemented to remove invalid survey entries: (1) duplicate IP addresses, email addresses, phone numbers and/or other contact information; (2) unusually short duration of survey completion (<85 s); and/or (3) suspicious write-in responses deemed invalid by study staff (e.g., smoke 1000 cigarettes per day). Of the 801 people who were screened for participation, 287 (35.83%) were eligible and had valid responses based on the above data screening procedures. Participants who were eligible and completed the entire survey were compensated USD 2.

### 2.2. Measures

To prepare this survey, we consulted several COVID-19-related assessment resources, which were emerging as the pandemic began. These included: (1) survey items established through a trans-National Institutes of Health (NIH) workgroup led by the National Institute on Aging and the Office of Behavioral and Social Sciences Research and made available through the National Institutes of Health (NIH) Public Health Emergency and Disaster Research Response (DR2) and PhenX Toolkit repositories; and (2) survey items established through a FDA-funded Tobacco Centers for Regulatory Science COVID-19 and NIH Tobacco Regulatory Science Working Group.

*Self-reported medical status*. Participants indicated whether they had ever been told by a doctor or health professional that they have any medical conditions associated with increased risk of severe COVID-19 illness by indicating “yes or no” [37]. The specific conditions were as follows, which reflected the known pre-existing conditions at the time associated with risk of a more severe COVID-19 illness: lung disease (i.e., COPD/emphysema/chronic bronchitis, asthma, interstitial lung disease/pulmonary fibrosis), kidney disease, liver disease, obesity, diabetes, cancer, serious heart condition (i.e., heart failure, coronary heart disease, congenital heart disease, cardiomyopathies, pulmonary hypertension) and/or a compromised immune function. This question aligns with several self-reported chronic health condition single survey items in various PhenX Toolkit protocols. 

*COVID-19 risk perceptions.* Participants rated their perceptions of risk from COVID-19 via 2 items using a 7-point scale from 1 = “definitely no” to 7 = “definitely yes”: *How concerned about coronavirus (COVID-19) are you for your own health?* and *How concerned about coronavirus (COVID-19) are you for the health of others in your community?* They also rated how much they perceived that their smoking increased their risk of harm from COVID-19 on a 7-point scale from 1 = “definitely no” to 7 = “definitely yes”: *Do you think your tobacco use increases your risk of harm from coronavirus (COVID-19)?* These first three survey items were developed through the FDA-funded Tobacco Centers for Regulatory Science COVID-19 and NIH Tobacco Regulatory Science Working Group and have been published in two survey studies evaluating COVID-19 risk perceptions and changes in opiate use [38] and cigarette smoking and e-cigarette use [20].

Participants who self-reported any medical conditions above were asked an additional COVID-19 risk perception item using the same 7-point scale: *Are you concerned that these medical condition(s) increase your risk of harm from coronavirus (COVID-19)?* We developed the third survey item related to medical conditions from the other two items for the purpose of this study.

*Cigarette smoking use behaviors.* Participants completed the Heaviness of Smoking Index (HSI) [39], which measures the number of cigarettes smoked per day, the time to first cigarette and yields a total nicotine dependence score (i.e., ≥5 indicative of high dependence). Participants were then asked about the impact of the COVID-19 pandemic, from the start in March 2020 until the time of the survey, on their smoking: *We are interested in the impact that the coronavirus (COVID-19) pandemic has had on how much you have been smoking. Has your smoking stayed the same, decreased, increased, or other (please specify)?* This item was adapted from an item developed by the FDA-funded Tobacco Centers for Regulatory Science COVID-19 and NIH Tobacco Regulatory Science Working Group and published in two survey studies evaluating COVID-19 risk perceptions and substance use [20,38]. We coded smoking changes into a single categorical variable: (1) smoking decreased, (2) smoking increased, (3) smoking stayed the same.

Participants then selected the associated reasons for these changes. For no change in use, participants indicated one of the following reasons: (1) “enjoy using”; (2) “not concerned that use increases my risk of COVID-19;” (3) “do not see any reason to change use due to COVID-19;” (4) “other (please specify).” For increased use, participants selected one of the following: (1) “stress;” (2) “more free time/boredom;” (3) “staying home more (i.e., fewer restrictions on smoking);” “other (please specify).” For decreased use, participants indicated one of these reasons: (1) “health concerns due to COVID-19;” (2) “reduced ability to afford due to COVID-19;” (3) “difficulty accessing due to COVID-19 (i.e., store closures, avoiding public places, transportation issues);” (4) “other (please specify).”

Cigarette smoking related cognitions. They also reported whether the pandemic impacted their motivation to quit smoking: “How has your motivation to quit smoking changed since you learned about the coronavirus pandemic (COVID-19)?” by selecting “increased,” “decreased,” “stayed the same,” “other (please specify).” This item was selected from the FDA-funded Tobacco Centers for Regulatory Science COVID-19 and NIH Tobacco Regulatory Science Working Group measures and has been published in two survey studies [20,38]. We coded motivation changes into a single categorical variable: (1) motivation decreased, (2) motivation increased, (3) motivation stayed the same. 

### 2.3. Data Analysis

Data were analyzed using SPSS software (version 26). Descriptive statistics were used to summarize frequencies and percentages of sociodemographic characteristics (age, race, ethnicity, gender) and clinical characteristics (co-morbid medical conditions, COVID-19 risk perceptions related to smoking). Chi-square tests, *t*-tests and analyses of variance were used to compare associations among sociodemographic/clinical characteristics and cigarette smoking behaviors and cognitions. We then performed multinomial logistic regression, adjusted for sociodemographic characteristics, to evaluate whether clinical characteristics were associated with smoking change membership (i.e., no change, decreased smoking, or increased smoking). We used no change in smoking as the reference category. Multinomial logistic regression is an extension of binary logistic regression for modeling the probabilities of a nominal dependent outcome with more than 2 discrete outcomes.

Complete data were available for >95% of cases. Only one participant had missing data for the primary 3-level smoking change outcome variable, so we deleted this case for a final analytic sample of N = 286. The missing data for other explanatory variables are reported in Table 1 below.

## 3. Results

### 3.1. Sample Characteristics

The final enrolled sample (N = 286) was mostly non-Hispanic White (62.9%) and female (54.0%) (see Table 1 for a breakdown of participants by demographic characteristics). These sample demographic characteristics are representative of the most recent 2021 U.S. Census estimates, which reported that the U.S. population is 60.1% non-Hispanic White, 13.4% Black, 1.3% American Indian/Alaskan Native, 5.8% Asian, 0.2% Native Hawaiian/Other Pacific Islander, 2.8% More than one race, 18.5% Hispanic/Latino and 50.8% female [40]. Though our sample was slightly more non-Hispanic White and female compared with the Census figures, we had higher percentages of participants who identified as Black and Native Hawaiian/Other Pacific Islander. Participants had a mean age of 43.69 years (SD = 14.73). Approximately 8.4% of our sample was 65 years of age or older, which is consistent with the most recent estimate of 9% smoking prevalence in this demographic [41].

Most participants reported daily smoking (90.1%). They had a mean Heaviness of Smoking Index score of 2.81 (SD = 1.49), indicative of low to moderate nicotine dependence [39]. Half (51.7%) endorsed having a chronic medical condition associated with greater COVID-19 risk, with lung disease being the most commonly reported condition (25.9%). More than half of the sample indicated knowing someone who had COVID-19 (60.7%). Of this subsample, approximately two-thirds (67.6%) reported that a close friend or relative had COVID-19. Recent (i.e., past 90 days) alcohol use (63.6%) and other tobacco product use (52.1%) (i.e., e-cigarettes/vaping products, cigars, cigarillos, pipe tobacco, hookah) were common among the sample.

### 3.2. COVID-19 Risk Perceptions 

On average, participants reported being moderately concerned about COVID-19 for their own health (*M* = 5.26, *SD* = 1.80) and that of others (*M* = 5.62, *SD* = 1.47). Personal COVID-19 risk perceptions were greater among younger participants (*r* (281) = −0.24, *p* < 0.001); Hispanic/Latino (*M* = 5.90, *SD* = 1.45) versus non-Hispanic/Latino participants (*M* = 5.15, *SD* = 1.84; *t* (59.83) = −2.89, *p* = 0.003); Black (*M* = 6.08, *SD* = 1.38) versus White participants (*M* = 5.04, *SD* = 1.83); *F* (2, 282) = 7.35, *p* < 0.001). Participants with at least one co-morbid medical condition were moderately concerned, on average, that their condition could increase their risk of harm from COVID-19 (*M* = 5.52, *SD* = 1.72). 

### 3.3. COVID-19 Pandemic and Cigarette Smoking Changes

Participants were somewhat concerned that smoking increased their risk of harm from COVID-19 (*M* = 5.16, *SD* = 1.93), and more than a third reported that their motivation to quit smoking increased because of COVID-19 (40.1%; 109/286). Participants reported the following actual impact of the pandemic on their smoking behavior: 25.2% increased their smoking, 29.4% decreased their smoking, and 45.5% experienced no change. 

COVID-19 concerns were the main reason participants gave for decreased smoking (73.5%; 61/83), but some also reduced smoking because of financial reasons (27.7%; 23/83) and/or reduced access to cigarettes (37.3%; 31/83). Negative emotional states, such as stress (70.8%; 51/72) and boredom (63.9%; 46/72), were key explanations for increased smoking. Many also increased their smoking because of fewer restrictions on smoking due to staying home (56.9%; 41/72). The primary reason participants cited for not changing their smoking was that they liked smoking (64.6%; 84/130). Some did not change their smoking due to a lack of concern about smoking-related COVID-19 risks (20.8%; 27/130), limited perceived reasons to change (30.0%; 39/130), or difficulty stopping (13.1%; 17/130). 

Participants’ clinical characteristics (i.e., COVID-19 risk perceptions, self-reported co- medical status) were then evaluated for potential associations with smoking change membership (i.e., no change, decrease, increase) in multinomial logistic regression models adjusted for demographic variables (age, gender, race, ethnicity). As shown in Table 2 and Figure 1, the odds of decreased smoking (i.e., compared to the reference category of no smoking change) were greater among younger participants, those with higher COVID-19 risk perceptions and people who identified as Black compared to White (55.8% versus 21.3%). Conversely, the odds of increased smoking (i.e., compared to no smoking change) were greater among participants who reported having any co-morbid medical condition relative to those who did not report any medical conditions (65.3% versus 34.7%) (see Table 2 and Figure 2).

## 4. Discussion

The goal of this study was to further examine smoking behavior among U.S. adults in response to the COVID-19 pandemic with a particular focus on co-morbid medical conditions linked to increased COVID-19 risk. Most participants indicated no change in their smoking, but a number increased or decreased their use. Participants who smoked cigarettes were moderately concerned that their use increased their risk of harm from COVID-19. These concerns were associated with increased quit intentions to reduce smoking and were the primary reason participants reported for actual reductions in use. In models evaluating smoking change, endorsing greater COVID-19 risk perceptions was associated with higher odds of reducing compared to no change. Conversely, having at least one co-morbid medical condition was related to higher odds of increased smoking relative to no change. In addition, we found greater smoking reductions among younger and Black participants. Our findings are in line with prior COVID-19 research that showed greater COVID-19 risk perceptions increased motivation to quit and/or promoted reductions in smoking and e-cigarette use [20,29,30,31,32,42,43,44,45]. Likewise, the results are consistent with other studies conducted early during the pandemic, which found associations among tobacco use reductions and younger age [46,47,48] and Black race [31]. Beyond replicating the results from recent studies, our study adds to the evidence base on COVID-19 and smoking by characterizing the relation between medical status and smoking changes. 

It is a concerning finding that individuals with co-morbid medical conditions were more likely to increase their smoking given their higher risk of more severe COVID-19 illness [49]. One potential explanation for this association is that those with medical conditions might have felt more stress and/or experienced greater mental health complications because of their medical status, which, in turn, promoted smoking [25,50,51]. Another possibility is that having a medical co-morbidity may be related to greater difficulty quitting smoking. Findings from the International Tobacco Control (ITC) Smoking and Vaping Survey provide some support for this hypothesis. In two ITC studies, having a co-morbid health concern/condition was associated with lower smoking cessation self-efficacy, lower than expected cessation activities and use of evidence-based cessation treatments [52] and decreased cessation success among participants with some conditions (e.g., chronic lung diseases) [53]. In addition, participants with medical conditions in our study sample were significantly older than their healthy counterparts, and younger participants were more likely to reduce their smoking, which corresponds to other smoking prevalence data during the pandemic [48]. Older individuals who smoke may be less interested in quitting and/or experience greater difficulty quitting than younger individuals [54]. Though the exact mechanisms of these poorer smoking outcomes are unclear, it has been suggested that older smokers and/or those with co-morbid medical conditions might be more nicotine dependent or perceive less health benefit from smoking cessation [55,56,57]. Age was significantly associated with smoking changes in this sample, with younger age related to increased odds of reducing smoking. A final explanation for increased smoking among individuals with co-morbid medical conditions may be comparative optimism (i.e., the belief that one is less likely to experience a negative event compared to others), as suggested in a recent study of 2768 smokers’ perceived general and personal susceptibility to and seriousness of COVID-19 [58]. Specifically, Vogel and colleagues found increased smoking was associated with greater perceived susceptibility but lower perceived seriousness of COVID-19, which suggests that while many individuals may have believed that smoking increased COVID-19 risk, they perceived that their personal risk level was low [58].

### 4.1. Limitations

Important study limitations should be noted. We assessed health status and behavioral changes at a single time during the first wave of the COVID-19 pandemic. The cross-sectional design limits causal conclusions that can be drawn about the pandemic and smoking and how these associations may have evolved over time following subsequent waves of the illness. Assessments, though derived from expert panels as the pandemic unfolded, were limited to self-report. In addition, we utilized convenience sampling methods to recruit participants, which limits the generalizability of the results. However, our final enrolled sample closely approximated U.S. population demographics. 

### 4.2. Future Directions/Recommendations

These findings, in combination with recent research, have implications for tobacco treatment during the COVID-19 pandemic and other natural disasters/public health threats. Healthcare providers should do more to ensure that older smokers and those with co-morbid medical conditions receive the most effective cessation interventions as well as psychoeducation regarding the general and COVID-19-specific gains that can still be achieved from quitting. Given the substantial risks of severe and more prolonged illness in this subgroup, smoking cessation interventions that are better tailored to their unique needs (e.g., higher nicotine dependence, stress, limited perception of benefit from cessation) may be needed. Likewise, due to reduced healthcare visits during COVID-19 among this demographic [59], tobacco treatment adaptations may also be needed, such as the expansion of telehealth and greater use of targeted outreach strategies [60]. However, this tobacco cessation support should not be limited to these special populations. As preventive healthcare visits have reduced during the ongoing COVID-19 pandemic [61], any patient’s self-report of smoking should be addressed in any health context in which care is being provided [62]. Additionally, healthcare providers and tobacco interventions should tailor care to better address patients’ needs during the unique circumstances of the COVID-19 pandemic, such as use of smoking as a primary coping response, misperceptions regarding personal susceptibility to COVID-19 and/or smoking risks and practical barriers to behavior change and treatment engagement, which may be exacerbated during this time. More research is also needed on effective public health messages [63] for promoting smoking cessation and counteracting potential misinformation during COVID-19 and other pandemics, particularly for individuals who may be at greatest risk of illness. Future studies should also explore smoking and prolonged COVID illness and effective strategies for promoting smoking cessation in these patients. Lastly, to better understand and support smoking behavior change among adults with co-morbid medical conditions during the ongoing COVID-19 pandemic, molecular biology data should be combined with other research methods.

## 5. Conclusions

Following the first wave of the COVID-19 pandemic in the U.S., our survey results suggest that individuals who smoked cigarettes reported varied changes in their use. Most people did not change their use. However, greater perceived risks from COVID-19 appeared to promote reductions in cigarette use for some, whereas having a co-morbid medical condition may have increased smoking for others.

## Figures and Tables

**Figure 1 ijerph-19-08239-f001:**
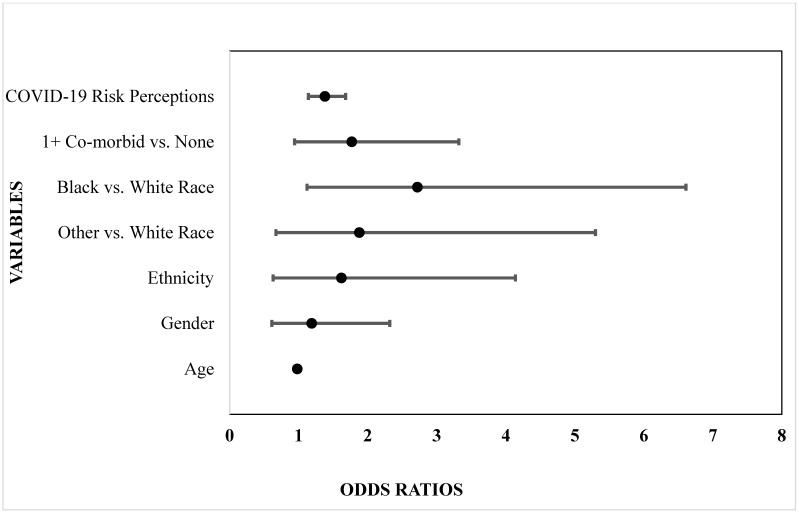
Smoking decreased relative to no change.

**Figure 2 ijerph-19-08239-f002:**
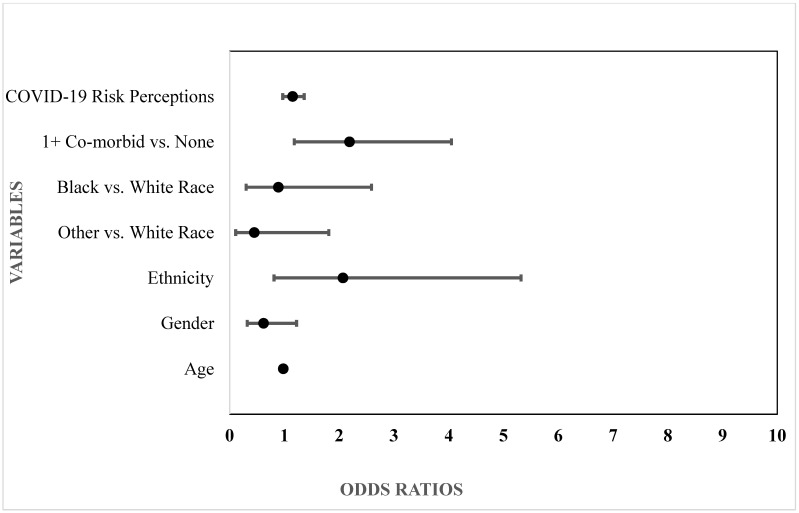
Smoking increased relative to no change.

**Table 1 ijerph-19-08239-t001:** Participant self-reported characteristics (N = 286).

Characteristic	Categories	N	%
**Race**	American Indian/Native Alaskan	8	2.8
	Asian	5	1.7
	Black	52	18.2
	More than 1 race	4	1.4
	Native Hawaiian/Pacific Islander	6	2.1
	White	210	73.4
	Not reported/unknown	1	0.3
**Ethnicity**	Hispanic or Latino	40	14
	Not Hispanic or Latino	243	85
	Not reported/unknown	3	1
**Gender**	Female	154	53.8
	Male	128	44.8
	Non-binary	1	0.3
	Transgender	2	0.7
	Not reported/unknown	1	0.3
**Age**	18–30	76	26.6
	31–40	56	19.6
	41–50	50	17.5
	51–60	62	21.7
	61 and older	41	14.3
	Not reported/unknown	1	0.3
**Cigarettes per day**	10 or less	120	42
	11–20	100	35
	21–30	46	16.1
	31 or more	17	5.9
	Not reported/unknown	3	1
**Co-morbid Medical Condition**	Any	148	51.7
	Cancer	12	4.2
	Cardiovascular Disease	23	8
	Compromised Immune Function	14	4.9
	Diabetes	30	10.5
	Kidney Disease	12	4.2
	Liver Disease	6	2.1
	Lung Disease	74	25.9
	Obesity	49	17.1
**Know someone who had COVID-19**	No	112	39.2
	Yes	173	60.5
	Not reported/unknown	1	0.3

**Table 2 ijerph-19-08239-t002:** Multinomial logistic model of participant clinical characteristics associated with odds of decreased or increased cigarette smoking compared to no smoking change in response to the COVID-19 pandemic adjusted for demographic variables.

Variables	Smoking Decreased				Smoking Increased			
	*B* (SE)	OR (95% CI)	Test Statistic *Wald (df)*	*p*	*B* (SE)	OR (95% CI)	Test Statistic *Wald (df)*	*p*
COVID-19 Risk Perceptions	0.32 (0.10)	1.38 (1.14–1.68)	10.79 (1)	0.001	0.14 (0.09)	1.15 (0.97–1.36)	2.42	0.12
Co-morbid Medical Condition	0.57 (0.32)	1.77 (0.94–3.32)	3.17 (1)	0.08	0.78 (0.32)	2.19 (1.18–4.05)	6.15	0.01
Race								
Black	1.00 (0.45)	2.72 (1.12–6.61)	4.86 (1)	0.03	−0.12 (0.55)	0.89 (0.30–2.59)	0.05 (1)	0.83
Other	0.63 (0.53)	1.88 (0.67–5.30)	1.42 (1)	0.23	−0.81 (0.72)	0.45 (0.11–1.81)	1.28 (1)	0.26
White	Reference	---	---	---	Reference	---	---	---
Ethnicity	0.48 (0.48)	1.62 (0.63–4.14)	1.01 (1)	0.32	0.73 (0.48)	2.07 (0.81–5.32)	2.28 (1)	0.13
Gender	0.17 (0.34)	1.19 (0.61–2.32)	0.26 (1)	0.61	−0.47 (0.34)	0.62 (0.32–1.22)	1.89 (1)	0.17
Age	−0.03 (0.13)	0.98 (0.95–1.00)	4.75 (1)	0.03	−0.02 (0.01)	0.98 (0.96–1.01)	1.84 (1)	0.18

Note. Ethnicity scored 1 = non-Hispanic, 0 = Hispanic. Gender scored 1 = male, 0 = female (3 other values coded as missing due to model convergence warning with this variable). Co-morbid Medical Condition scored 1 = no conditions, 0 = 1 or more conditions. Race scored 2 = White, 1 = Black, 0 = Other. Smoking outcome scored 2 = no change, 1 = decrease, 0 = increase. For analyses, the reference category for variables and the outcome was the highest value (i.e., 1 or 2).

## Data Availability

The data underlying this article will be shared on reasonable request to the corresponding author.

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
