# Peer review of "Cigarette Smoking in Response to COVID-19: Examining Co-Morbid Medical Conditions and Risk Perceptions"

_ijerph, 2022, doi:10.3390/ijerph19148239_

Round 1

Reviewer 1 Report

None. Clear and interesting study. 

Reviewer 2 Report

Tobacco usage has been linked to several health problems. Concerning the COVID-19 pandemic, inquiries have been made on the clinical effects for smokers, including whether or not they are just as vulnerable to infection and whether or not nicotine has any biological impact on the SAR-CoV-2 virus. This is an interesting topic of research and authors have touched the base regarding participants’ smoking behavior in response to the COVID-19 pandemic as well as possible correlates of use including demographic characteristics, co-morbid medical conditions, and COVID-19 risk perceptions.

For the future, I would suggest the authors to incorporate molecular biology data in combination with survey questionnaire which will improve the overall methodology and a concrete output of hypothesis in question. The conclusions are in the line with study question. All the references are relevant and consistent with the text.

Although the authors have accumulated a substantial quantity of data, in my opinion, it should have been presented in a more compelling manner. The readability of the manuscript might be improved with the addition of graphical illustrations.

Reviewer 3 Report

Comments to authors

This article is focused on “Cigarette Smoking in Response to COVID-19: Examining CoMorbid Medical Conditions and Risk Perceptions.” In this Ms, Authors addresses about the impact of COVID-19 on cigarette smoking in adults who reported recent use, with a particular focus on chronic medical conditions likely associated with increased COVID-19 risk. This study is significant for the society and research community. Authors must incorporate replies to the comments.

Comments

1.      Authors should need to strengthen the introduction section.

2.      Authors should add how such survey can be beneficial to the society.

3.      Authors can illustrate their study in a flowchart (in Materials and Methods section).

4.      Authors should briefly discuss about the Multinomial Logistic Model, and how it is significant in Material an methods section?

5.      Few more latest relevant reference should be incorporated in Ms.

6.      Future directions and recommendations should be incorporated in Conclusion section.

7.      Authors must cross check references from text to reference list, vice versa.

8.      Authors must check any typological and grammatical errors to avoid any mistake.

Author Response

Please see attached letter.
